# Leaf Spectral Analysis for Detection and Differentiation of Three Major Rice Diseases in the Philippines

Jean Rochielle F. Mirandilla [1,2,*], Megumi Yamashita [2], Mitsunori Yoshimura [3] and Enrico C. Paringit [4]

1   Philippine Rice Research Institute, Science City of Munoz 3119, Philippines
2   Graduate School of Agriculture, Tokyo University of Agriculture and Technology, 3-5-8 Saiwai-cho, Fuchu, Tokyo 183-8509, Japan; meguyama@cc.tuat.ac.jp
3   Department of Forest Science, College of Bioresource Sciences, Nihon University, 1866, Kameino, Fujisawa 252-0880, Japan; yoshimura.mitsunori@nihon-u.ac.jp
4   Department of Science and Technology, Philippine Council for Industry, Energy and Emerging Technologies R&D (DOST-PCIEERD), University of the Philippines Diliman, Quezon City 1101, Philippines; ecparingit@up.edu.ph
*   Correspondence: jrfmirandilla@philrice.gov.ph or s235342z@st.go.tuat.ac.jp

**Abstract:** Monitoring the plant's health and early detection of disease are essential to facilitate effective management, decrease disease spread, and minimize yield loss. Spectroscopic techniques in remote sensing offer less laborious methods and high spatiotemporal scale to monitor diseases in crops. Spectral measurements during the development of disease infection may reveal differences among diseases and determine the stage it can be effectively detected. In this study, spectral analysis was performed over the visible and near-infrared (400–850 nm) portions of the spectrum to detect and differentiate three major rice diseases in the Philippines, namely tungro, BLB, and blast disease. Reflectance of infected rice leaves was recorded repeatedly from inoculation to the late stage of each disease. Results show that spectral reflectance is characteristically affected by each disease, resulting in different spectral, signature sensitivity, and first-order derivatives. Red and red-edge wavelength ranges are the most sensitive to the three diseases. Near-infrared wavelengths decreased as tungro and blast diseases progressed. In addition, the spectral reflectance was resampled to common reflectance sensitivity bands of optical sensors and used in the cluster analysis. It showed that BLB and blast can be detected in the early disease stage on the IRRI Standard Evaluation System (SES) scale of 1 and 3, respectively. Alternatively, tungro was detected in its later stage, with an 11–30% height reduction and no distinct yellow to yellow-orange discoloration (5 SES scale). Three regression techniques, Partial Least Square, Random Forest, and Support Vector Regression were performed separately on each disease to develop models predicting its severity. The validation results of the PLSR and SVR models in tungro and blast show accuracy levels that are promising to be used in estimating the severity of the disease in leaves while RFR shows the best results for BLB. Early disease detection and regression models from spectral measurements and analysis for disease severity estimation can help in disease monitoring and proper disease management implementation.

**Keywords:** spectral reflectance analysis; rice foliar diseases; early detection; differentiation

## 1. Introduction

The Philippines is one of the major rice-consuming nations in Asia and the world. Rice is the most important staple and agricultural crop among Filipinos, accounting for 37% of the average daily diet and its industry employs 2.5 million households [1,2]. According to the Philippine Statistics Authority, 4.8 M ha of palay were planted in 2017, and the average yield of palay was 4 tons/ha, 4.41 tons/ha in irrigated and 3.11 in rainfed rice areas [3]. Being one of the major rice-consuming nations in Asia and the world, the Philippines has been putting in efforts to achieve rice self-sufficiency for decades and

increase competitiveness with other countries. One of the ways to improve rice production is to minimize yield losses from abiotic (i.e., climate change) and biotic stresses (i.e., pests).

Pests and their damage are mainly responsible for crop losses. The losses in crop yield due to pathogen infections, animals, and weeds in rice are at 30% globally [4]. Pest problems are also common in rice production in the Philippines. Rice tungro, bacterial leaf blight (BLB), and fungal diseases such as rice blast and sheath blight are among the major rice diseases affecting the country's rice fields [5]. For instance, in 2016, the Philippines' Department of Agriculture reported that rice blast and sheath blight threatened rice production on Panay Island. Around 3155 hectares were already affected by the rice blast in the provinces of Iloilo, Aklan, Capiz, and Antique [6]. The management of these diseases focuses on prevention, proper detection, and adequate control through good cultural management practices such as planting seeds of good quality, proper water management and fertilizer application, and good sanitation in the field.

Monitoring and early detection of these diseases are necessary to minimize these losses on agricultural production, reduce disease prevalence, and facilitate effective management practices. Information about the incidence and severity of the pest damage is particularly important for making rapid and timely management decisions to avert yield loss. Conventionally, diseases are detected by looking at symptoms. However, visual inspection is often laborious, time-consuming, and hard to estimate in large-scale farming. Other detection methods are expensive and require high-end laboratories, such as in conventional pathotyping, as well as several other novel genetic and serological tools, such as monoclonal antibodies and PCR-based markers, which are effective in the detection and analysis of the serological and genetic diversity of the pathogen population [7–10].

Remote sensing, an indirect method, is a less laborious and high-spatiotemporal-scale technique showing the possibility of detecting crop diseases. Spectroscopic and imaging techniques assume that stress affects the physical structure and photosynthesis of plants, influencing light energy absorption and reflectance [10,11]. Spectroscopic techniques include the measurement and analysis of measurements from contiguous spectral bands describing the interaction between a source of the electromagnetic radiation and the target. The advantages of these data include (1) unique spectral reflectance based on the plant's biochemical and physiological properties that give accurate identification and mapping of the targeted plant, and (2) quantitative estimates of biophysical absorptions that can be used to improve the understanding of ecosystem functioning and properties [12]. The utility of remote spectral data to diagnose crop pests (i.e., diseases and insects) can improve detection speed and provide an opportunity for non-destructive sampling [12,13]. Several studies have been conducted to assess the use of reflectance data in the detection of various diseases in rice, such as panicle blast, bacterial leaf blight (BLB), brown spot, and injuries from brown planthopper (BPH) in a certain rice stage [14–17]. These pests and diseases exhibit different symptoms, levels of incidence, and yield reductions in rice. The studies used airborne hyperspectral imagery [16] and ground-based hyperspectral sensors [18] to monitor rice diseases and severity, such as panicle blast and BLB. Different wavebands and band ratios were significantly correlated with the disease incidence and could discriminate between healthy and diseased plants [16,18,19].

Although these research studies were established to prove that spectral reflectance can detect diseases in rice, they usually focused on a certain stage of the disease infection for measurement and on comparing severely diseased plants and healthy plants. There is a need to assess the effect of the disease, disease stages, and severity on spectral reflectance on rice as factors that must be considered when determining sensor sensitivity in disease detection and monitoring. Furthermore, to date, limited studies have been carried out to differentiate different foliar diseases in rice. Conventionally, measurements were performed only at a specific disease stage. However, time series measurements present the opportunity to assess the physiological symptoms of rice foliar diseases that influence changes in spectral signatures. Obtaining measurements at the different stages of disease infection can reveal the differences among these diseases and determine the stage at which they

can be effectively detected. Moreover, variations in spectral reflectance can be a basis for future designs of optical sensors to detect and differentiate plant diseases. Identifying the spectral bands and features that are sensitive to certain diseases is important for the effective application of remote sensing data analysis techniques.

This study aims to detect and differentiate three major diseases, i.e., blast, BLB, and rice tungro in the Philippines by performing a multi-temporal spectral analysis. The differences in spectral responses and affected spectral regions were identified to differentiate these diseases. Moreover, the potential of satellite and airborne optical sensors (e.g., Sentinel 2 and Parrot Sequoia) for early detection and discrimination was evaluated using these spectral features. Early detection of these major diseases will help to prevent possible disease outbreaks and yield loss due to severe infection. Properly identifying them will help implement appropriate and timely disease management alternatives.

## 2. Materials and Methods

### 2.1. Study Establishment

The study was established in the experimental field of the Philippine Rice Research Institute Central Experiment Station in the Science City of Muñoz, Nueva Ecija, Philippines (15.6713°N 120.8919°E) during the 2018 rice growing dry season (January to March). Both blast and tungro set-ups were planted inside the screen house to prevent the disease from spreading beyond the experimental area. Natural infection is also highly favorable within screen houses with the existing National Cooperative Testing (NCT) disease screening project. On the other hand, BLB was established in the experimental field as there is an existing set-up for disease screening within the area. The rice plants were established on different dates to ensure that the data for each disease will be gathered during the same period to minimize errors from varying illumination.

Two rice lines were planted in 4 m$^2$ to monitor the disease progress. There are three replications for both inoculated and uninoculated test entries which served as control (Table 1). The layouts for each disease can be seen in Figure 1; tungro (Figure 1a), BLB (Figure 1b), and blast (Figure 1c). The study used the NCT procedure for disease inoculation: tungro inoculation (Figure 1d), BLB clipping method (Figure 1e), and blast spreader rows (Figure 1f). Rice plants were inoculated during the most critical stage of the rice plant for the disease infection. Blast and tungro were inoculated at the seedling stage while BLB was inoculated at the maximum tillering stage. The crop management used during the experiment was based on the NCT protocols [20]. The IRRI Standard Evaluation System (SES) for rice [21] was used for visual assessment to determine the disease's degree of severity and incidence.

**Table 1.** List of lines used and establishment dates of each disease.

| Disease | Lines Used | | Transplanting Date | Inoculation Date |
| | Susceptible | Resistant | | |
| --- | --- | --- | --- | --- |
| Tungro | Taichung Native 1 (TN1) | IRGC21473 (ARC) | 10 January 2018 | 9 January 2018 |
| BLB | TN1 | IRBB57 | 1 February 2018 | 23 March 2018 |
| Blast | Malay 2 | CO39 | 21 March 2018 | 12 March 2018 |

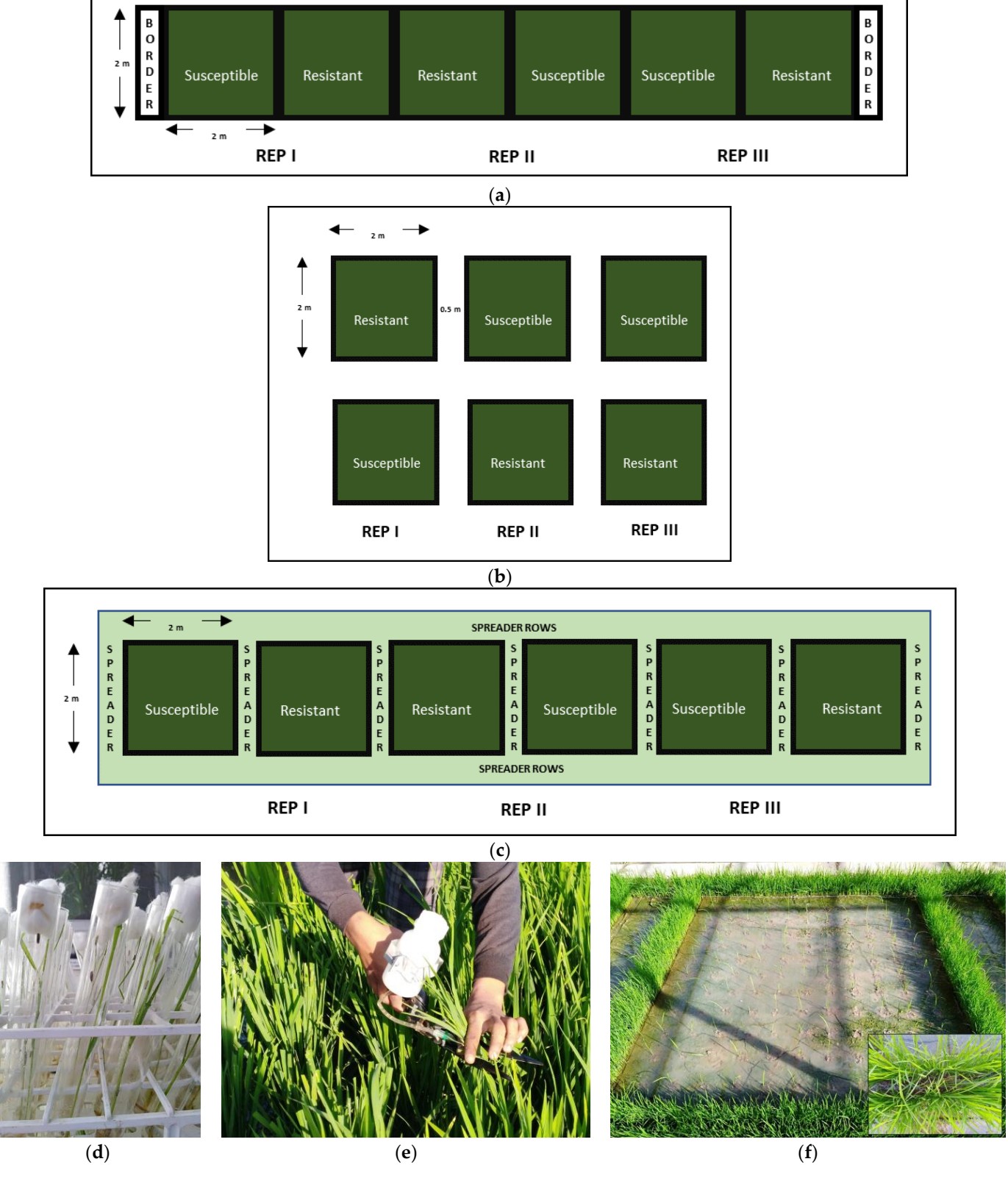

**Figure 1.** Experiment layout of (**a**) tungro in the screen house, (**b**) BLB in the experimental field, and (**c**) blast in screenhouse. Disease inoculation of (**d**) tungro, (**e**) BLB—clipping method, and (**f**) blast—spreader rows (inset—zoom picture of the spreader rows).

### 2.2. Data Gathering and Measurements

Spectral responses were measured using Ocean Optics USB4000-VIS-NIR-ES spectrometer (Ocean Optics Asia) and Spectralon® (Labsphere, Inc., North Sutton, NH, USA) (Figure 2a) almost every day between 10:00 a.m. and 2:00 p.m. ensuring high solar radiation except for cloudy and rainy days. Measurements were taken before and after disease inoculation, until the disease reached the score of 9 in SES, 38, 20, and 37 days for tungro, BLB, and blast, respectively. The spectrometer was set up at least 15 min before going to the field to make the instrument warm up to decrease errors associated with the detector. Sufficient illumination was ensured to obtain optimal results. Five plants per plot and replication were measured for the spectral responses.

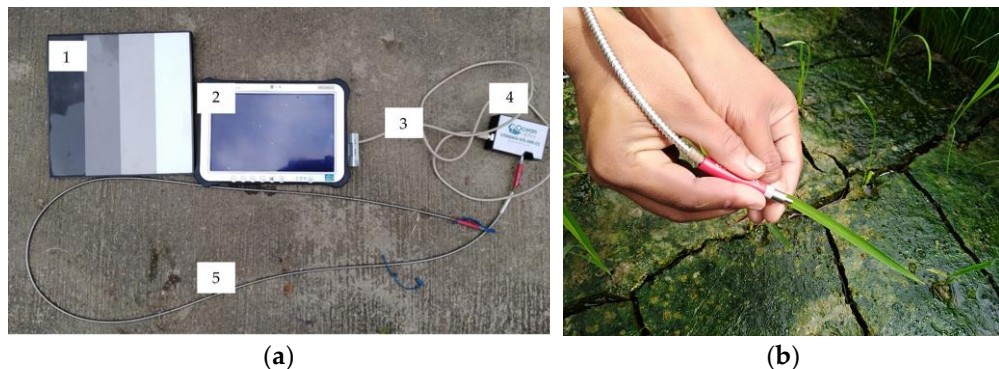

|(**a**)|(**b**)|

**Figure 2.** (**a**) Spectrometer set-up including the following instruments: (1) Spectralon, (2) Panasonic Toughpad, (3) USB interface cable, (4) USB4000-VIS-NIR-ES spectrometer, and (5) fiber optics used in the experiment; (**b**) spectral reflectance measurement.

One side of the optic cable was attached to the spectrometer, while the other side was mounted on a stand oriented at a certain degree from the sun. The fiber optics tip was placed on top of the white reference panel (99% reflectance) and the leaf. For dark object measurement, the tip was closed and covered with thick black cloth to ensure measurement of less than 10% reflectance. Then, the height of the tip of the fiber optic cable was set at a distance, ensuring that the tip's instantaneous field of view (IFOV) is smaller than the material being observed (Figure 2b). The targets should not have a shadow cast on them. A total of sixty measurements were carried out for each plot (4 leaves with 15 scans per leaf). All measurements were carried out in the same manner, in front of the leaves. A high number of scans can provide sufficient redundancy and robustness against response changes. Scans were saved in a text file extension.

Agronomic parameters (i.e., Leaf Area Index and plant height) and weather data were also collected once a week. Leaf Area Index (LAI) was measured using an Accu-PAR L P-80 ceptometer (Meter Group Inc., Pullman, WA, USA) in three replicates within each plot with at least two plants away from each other and bunds. Plant height was measured in five sampling plants per plot. The leaves of a single hill were gathered and measured vertically from the soil surface up to the end point of the longest leaf.

### 2.3. Data Preparation

All the saved text files containing the nominal radiance and irradiance were processed. The reflectance of each target object was computed from the average readings using

$$R(\lambda) = \frac{E(\lambda)_{sample} - E(\lambda)_{dark}}{E(\lambda)_{white} - E(\lambda)_{dark}} \tag{1}$$

where $E(\lambda)_{sample}$ is the nominal radiance, $E(\lambda)_{white}$ irradiance measured by the white panel, and $E(\lambda)_{dark}$ by the dark object.

The Savitzky–Golay smoothing method with five weighing coefficients was used to smoothen the spectral data. It is a digital filter that can be applied to a set of digital data points to increase the signal-to-noise ratio without appreciably distorting the signal. The wavelengths from 400 nm to 850 nm were used in the analysis due to high noise.

As the leaf spectra of tungro, blast, and BLB were collected from different cultivars, a spectral normalization was implemented at first to adjust the spectral data from different groups to an identical baseline. This procedure facilitates spectral comparison between different stressors [22], which suppresses illumination differences. The formula used was as follows:

$$R(\lambda)_{norm} = \frac{Ref(\lambda)_i}{\frac{1}{n}(\sum_{i=1}^{n} Ref(\lambda)_i)} \tag{2}$$

where $R(\lambda)_{norm}$ is the normalized reflectance for band $i$; $Ref(\lambda)_i$ is the original reflectance of the band; $n$ is the total effective number of bands.

First-order derivatives were computed to see the changes in spectral reflectance or radiance with respect to wavelength. It provides information on the rate of change, which is the slope, with respect to wavelength [23]. Moreover, ratio spectra (sensitivity) are a way to enhance differences between spectral signatures and determine sensitive and significant wavelengths for disease [24]. The ratio curve reflects both the change direction (increase or decrease) and the change magnitude of reflectance; it can be treated as a spectral signature of a specific stressor. The formula is as follows:

$$Ratio_{(stress)} = \frac{Ref_{(unhealthy)}}{Ref_{(healthy)}} \tag{3}$$

where $Ref_{(unhealthy)}$ is the average reflectance of the stressed samples; $Ref_{(healthy)}$ is the average reflectance of corresponding healthy samples.

Using ENVI 4.7 (Harris Geospatial Solutions, Inc., Boulder, CO, USA) spectral library builder, all the spectral measurements and computed spectral analysis were databased in different spectral libraries. The spectral libraries were saved as 8-bit grayscale images for better visualization of the reflectance trend (temporal trend). In addition, spectral libraries were resampled to different optical sensors (e.g., Sentinel 2A and 2B, Parrot Sequoia). Resampling is used to match the response of a known instrument to the wavelengths of a specific image input file. This will attempt to possibly detect these diseases using spectral reflectance from leaves. Sentinel 2 has 13 spectral bands from the visible, near infrared, and shortwave infrared. It includes three bands in the 'red edge', which can provide key information on the vegetation state. Based on the latest release (June 2022) of Sentinel 2A and 2B spectral function, the central wavelengths have slightly changed for the B02 band of S2A and S2B, and B01 of S2A, along with slight changes in the Full Width Half Maximum (FMWH) for most of the bands. On the other hand, Parrot Sequoia is a multispectral camera with four bands that can be mounted on an Unmanned Aerial Vehicle (UAV). The bands are centered in red, green, red-edge, and NIR, which are the most sensitive ranges in stress or diseases.

### 2.4. Statistical Analysis: Cluster Analysis, Partial Least Square Analysis, and Machine Learning Algorithms

Cluster analysis was performed using the resampled spectral response to Sentinel 2A, 2B, and Sequoia. This was to detect the possible earliest date of disease infection by grouping the dates with similar spectral values. Cluster analysis is an unsupervised classification technique to group similar observations into a number of clusters based on the observed values of several variables for each individual. It has known capabilities for differentiating between relevant and irrelevant variables; thus, the choice of variables included in a cluster analysis must be underpinned by conceptual considerations. In cluster analysis, hierarchical clustering is a tree-based representation of the observations, which is called a dendrogram [25]. The study used the agglomerative method or Agglomerative

Nesting (AGNES), which works in a bottom-up manner. The two 'closest' (most similar) clusters are then combined, conducted repeatedly until all subjects are in one cluster.

Different regression analyses were performed to detect the predictability of the three diseases: Partial Least Square Regression (PLSR), Random Forest Regression (RFR), and Support Vector Regression (SVR). All the statistical analyses were performed in R (version 4.2.3) using the packages pls [26], randomForest [27], and e1071 [28]. A total of 264 observations were used in all the regression analyses (Table 2). To develop the models, this multitemporal spectral reflectance dataset was divided into 70% training and 30% test datasets.

**Table 2.** The number of observations used in the different regression analyses.

| Treatment | Replications | Samples | Dates | Total |
|---|---|---|---|---|
| Susceptible | 3 | 3 | 11 | 99 |
| Resistant | 3 | 3 | 11 | 99 |
| Control Susceptible | 3 | 1 | 11 | 33 |
| Control Resistant | 3 | 1 | 11 | 33 |
| Total | | | | 264 |

PLSR with leave-one-out cross-validation was applied to the spectral reflectance of the three foliar diseases to develop models for estimating disease severity. It is a generalization of multiple linear regression that is used to build predictive models with multicollinear variables [26]. It can capture maximum variations associated with the spectra and a large number of descriptor variables with lower correlation [24,26], thus making it advantageous to use in spectral analysis and modeling of hyperspectral data. The optimal number of components used was determined by using one-sigma heuristic model in the pls package in R. It is an approach based on the standard error of the cross-validation residuals. The coefficient of determination ($R^2$) and relative root mean square mean (RMSE) are used to evaluate the accuracy of the PLSR model.

Random Forest Regression is a straightforward and highly accurate machine learning algorithm that calculates the average prediction of multiple decision trees. It can handle the data set containing continuous variables [29]. In this study, the ntree parameter was optimized (from 100 to 1500) based on the root mean square error of the training model while maintaining the mtry default value of 4.

On the other hand, SVR uses the principles of Support Vector Machine (SVM) algorithms to examine the linear relationship between two continuous variables. It works well in high-dimensional datasets such as multi-temporal data [29]. The hyperparameter (i.e., cost and gamma) tuning was performed in the e1071 package framework. The accuracy of the models for detecting the severity of the disease was evaluated using $R^2$ and RMSE.

## 3. Results

### 3.1. Disease Progression in the Leaf

The study used spectral data to identify bands or spectral features that are sensitive to the three diseases. Spectral data were also resampled to optical sensors to detect and differentiate the foliar diseases. The spectral measurements were carried out daily except for days that were too cloudy and rainy until the SES scale reached 9. Eleven dates were selected out of 20, 17, and 21 measurement dates for tungro, BLB, and blast, respectively. These were chosen based on their corresponding SES scale, which showed the appearance of the visual symptoms in the leaves of the plants (Table 3).

**Table 3.** Different dates of spectral measurement with corresponding SES scale score.

| Selected Date Number | SES Scale | Tungro | | BLB | | Blast | |
|---|---|---|---|---|---|---|---|
| | | Date [1] | DAI [2] | Date [1] | DAI [2] | Date [1] | DAI [2] |
| 1 | 0 | 18 January 2018 | 9 | 16 March 2018 | 0 | 25 March 2018 | 4 |
| 2 | 1 | 24 January 2018 | 15 | 26 March 2018 | 3 | 28 March 2018 | 7 |
| 3 | 1 | 25 January 2018 | 16 | 27 March 2018 | 4 | 29 March 2018 | 8 |
| 4 | 3 | 28 January 2018 | 19 | 28 March 2018 | 5 | 30 March 2018 | 9 |
| 5 | 3 | 31 January 2018 | 22 | 29 March 2018 | 6 | 3 April 2018 | 13 |
| 6 | 5 | 1 February 2018 | 23 | 30 March 2018 | 7 | 5 April 2018 | 15 |
| 7 | 5 | 3 February 2018 | 25 | 3 April 2018 | 11 | 6 April 2018 | 16 |
| 8 | 7 | 4 February 2018 | 26 | 4 April 2018 | 12 | 12 April 2018 | 22 |
| 9 | 7 | 9 February 2018 | 31 | 5 April 2018 | 13 | 18 April 2018 | 28 |
| 10 | 9 | 14 February 2018 | 36 | 6 April 2018 | 14 | 25 April 2018 | 35 |
| 11 | 9 | 16 February 2018 | 38 | 12 April 18 | 20 | 27 April 18 | 37 |

[1] Date: mm/dd/yy; [2] DAI: day after inoculation.

Leaf rolling started to show in the tungro-inoculated leaves 9 days after inoculation (DAI). The yellowing, which progressed to an orange discoloration of leaves, started at 15 DAI. Disease severity increased to 10% of the area in 15 DAI and reached the SES scale of 9 at 36 DAI when the diseased area exceeded 50% infection. The reduction in height was greater in the susceptible treatment than in other treatments (Figure 3).

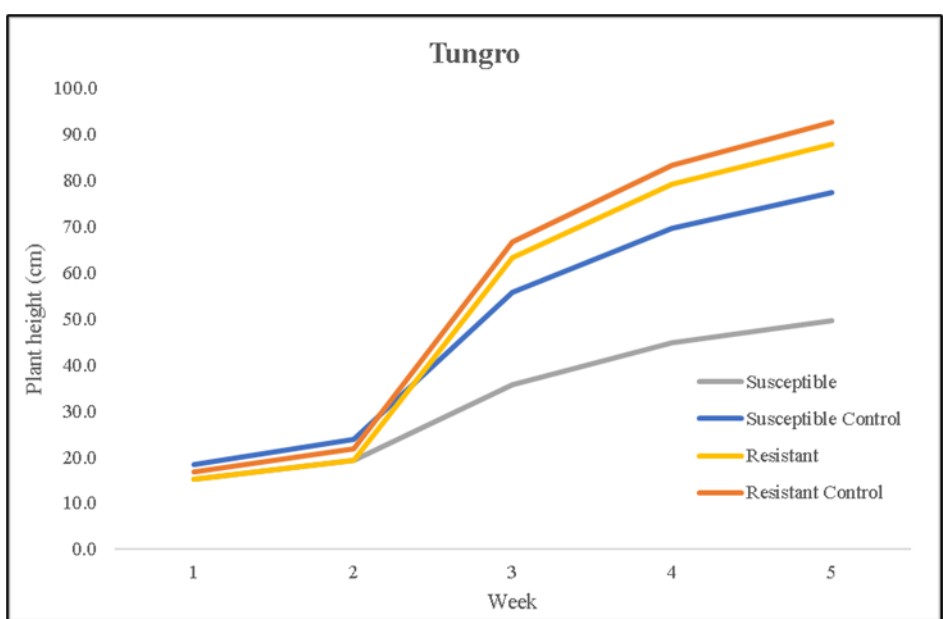

**Figure 3.** Plant height of different varieties in response to tungro infection.

Early symptoms of BLB infection were observed in the clipped tips at 3 DAI with an SES scale of 1 or 1–5% severity. After 20 days of inoculation, the scale reached SES 9 with more than 50% of the leaf area.

In the case of blast, the first symptom appeared in susceptible variety 7 DAI. Small brown specks of pin-point size or larger brown specks were observed in the leaves. The disease progressed to more than 75% leaf area affected in 37 DAI.

### 3.2. Differences in Multi-Temporal Spectral Reflectance among Tungro, BLB, and Blast

Inoculation and spectral measurements of rice plants for tungro were conducted only from the seedling to tillering stages. Tungro is most frequently seen in the vegetative phase (i.e., seedling and tillering stages); rice is most vulnerable at the tillering stage. Changes are noticeable in spectral reflectance signatures of leaves infected by tungro as the infection progresses (Figure 4a) in comparison to other treatments (Figure 4b–d). The normalized reflectance of the resistant and control treatments has the same trend from the start of measurements until the end. A small change can be observed in the reflectance ranges between 400 and 500 nm in the susceptible treatment. The normalized reflectance in 600–700 nm of the susceptible variety increased compared with the reflectance of the resistant and control treatments. In contrast, NIR region (700–850 nm) reflectance decreased, as observed in a slight color change in Figure 4e. The slope at the red-edge position (675–690 nm) between VIS and NIR became less steep and shifted to the left for susceptible treatment (Figure 4a). The shift can be seen from Date 1 to 11 as the color changes from red to a yellow shade (Figure 4e). Meanwhile, the other treatments showed no significant shift.

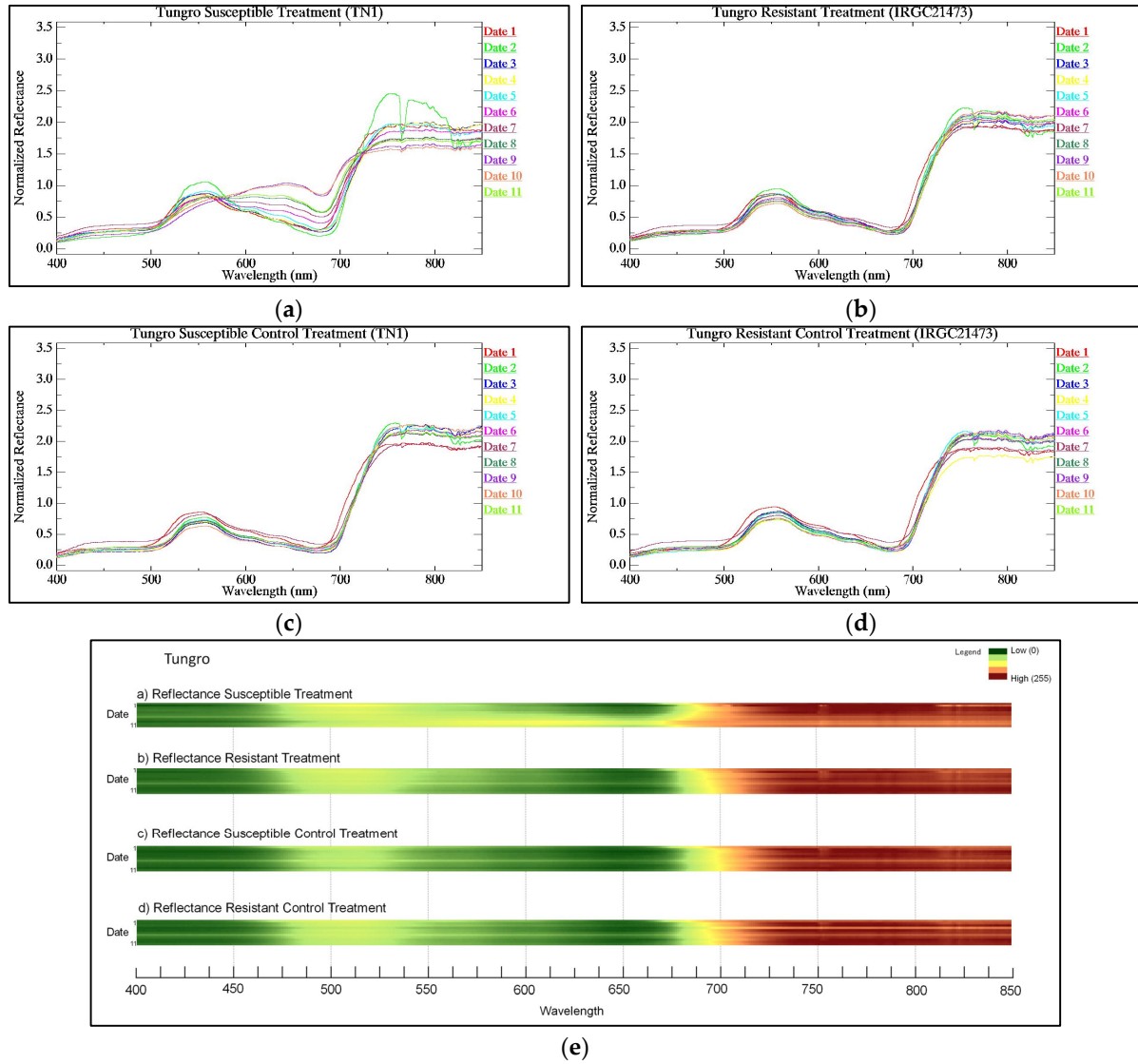

**Figure 4.** Leaf spectral reflectance among different varieties: (**a**) susceptible, (**b**) resistant, (**c**) susceptible control, and (**d**) resistant control in response to tungro infection; (**e**) 8-bit multitemporal spectral image of different tungro treatments.

Spectral measurements for BLB set-up started in early tillering until the flowering stage of the plant. Differences can be observed among the treatments inoculated by BLB and control, as shown in Figure 5. No significant change can be seen in the inoculated susceptible and control treatments (Figure 5b–d) except for the NIR range. A small decrease can be seen in this range as the rice plant matures. The infection showed no effect in the blue range wavelengths' (400–500 nm) reflectance in susceptible treatment until Date 11 (37 DAI). There was a significant increase in the inoculated susceptible treatment in 435–500 nm wavelengths when the disease reached 51–100% infection (Date 11). Furthermore, an increase in reflectance in the 550–680 nm wavelengths and a shift in the red-edge range (680–705 nm) in the susceptible variety were notable compared with other treatments (Figure 5e). The susceptible treatment also showed a distinct decrease in NIR reflectance in the susceptible variety from the first to the last date.

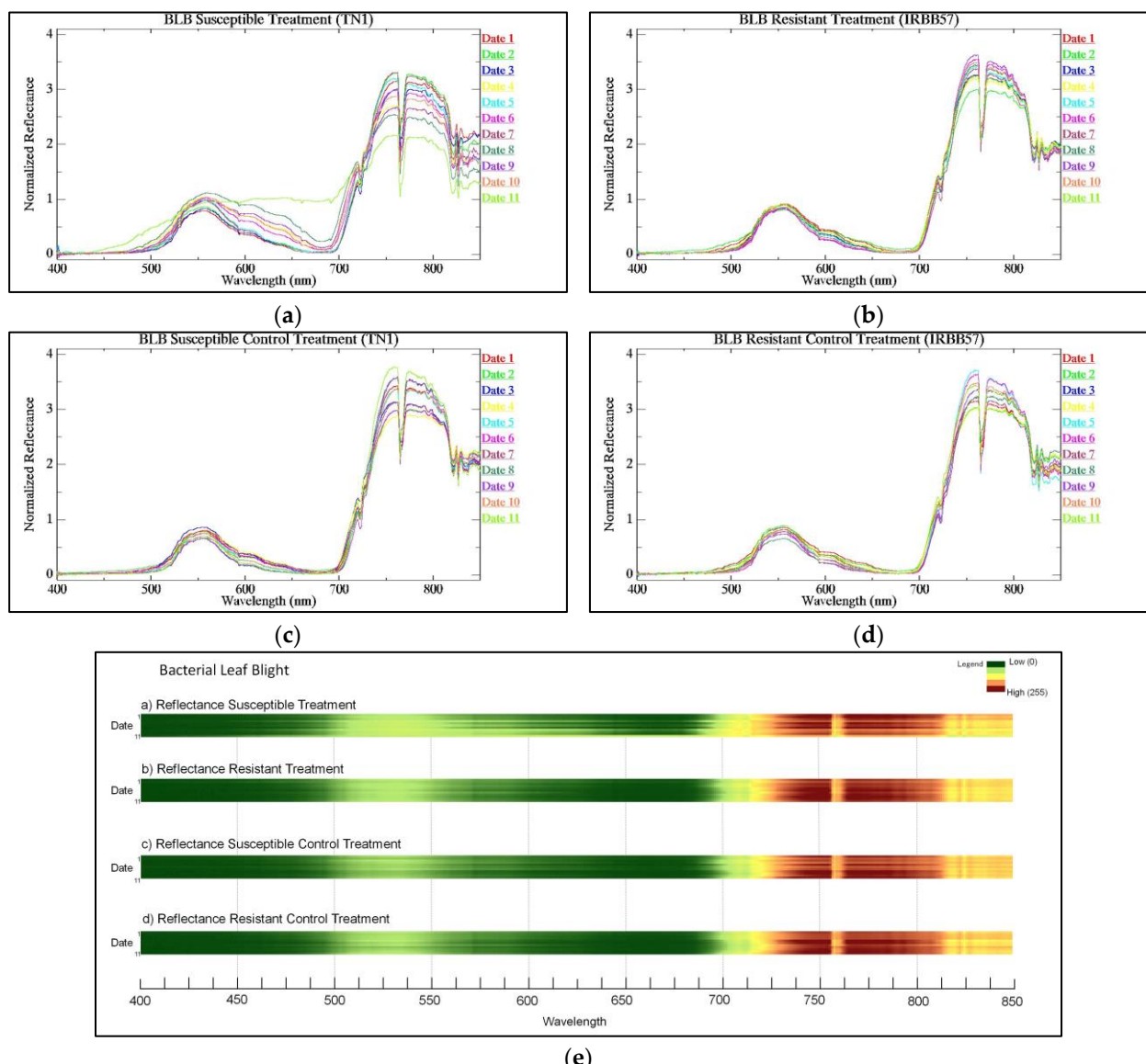

**Figure 5.** Leaf spectral reflectance among different varieties: (**a**) susceptible, (**b**) resistant, (**c**) susceptible control, and (**d**) resistant control in response to BLB infection; (**e**) 8-bit multitemporal spectral image of different BLB treatments.

Differences in reflectance spectra within the measurement dates were comparatively low for blast (Figure 6a) compared with the two other diseases (e.g., Figures 4a and 5a).

This can be due to its symptoms, as lesions initially appeared as pin-point-sized brown specks scattered in the leaves. The lesions became large, elliptical, or spindle-shaped with whitish to gray centers and red to brownish or necrotic borders that spread in the larger area of the leaf. Reflectance in the early days of the disease was a mixture of healthy and diseased parts of the leaves.

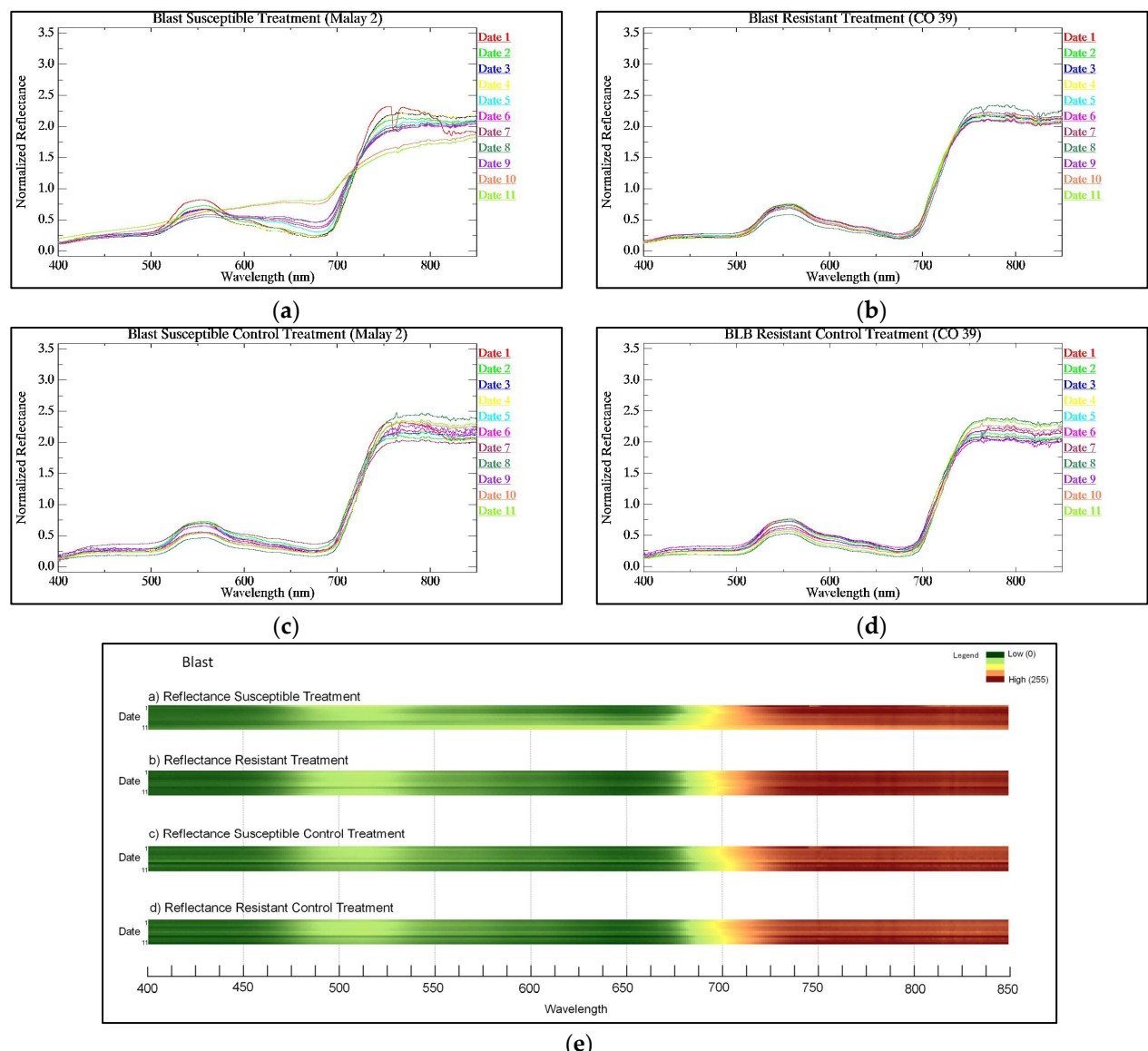

**Figure 6.** Leaf spectral reflectance among different varieties: (**a**) susceptible, (**b**) resistant, (**c**) susceptible control, and (**d**) resistant control in response to blast infection; (**e**) 8-bit multitemporal spectral image of different blast treatments.

No significant differences can be seen in the violet to the blue region (400–500 nm) among the treatments except in Date 11 of the susceptible treatment (Figure 6a). A significant change can be observed in the whole spectral reflectance in the last date, 37 DAI (Date 11). At this time, with 9-scale SES, the lesions spread in the leaves and started to wilt and die. Furthermore, the spectral reflectance in 600–700 nm wavelengths (i.e., red range) increased, while 720–850 nm (i.e., NIR range) decreased as the blast severity increased (Figure 6e). The red-edge (675–700 nm) slope shifted differently in the susceptible treatment compared to the other treatments; it shifted to the left while the other shifted to the right as the plants matured.

### 3.3. Comparison of Time-Series First-Order Derivatives and Sensitivity among Tungro, BLB, and Blast

Derivative spectroscopy uses changes in spectral reflectance or radiance with respect to wavelength to sharpen spectral features. Derivatives allow components of the spectrum to be more clearly separated.

Based on the first-order derivatives (Figure 7), there are differences between the susceptible variety and susceptible control among the three diseases. Tungro (Figure 7a) and blast (Figure 7e) susceptible treatments showed a shift in the wavelength ranges of 700–750 nm. The first-order derivative values decreased as the disease progressed in the red-edge to NIR ranges. The red-edge region covered the wavelength range between the red band absorption and the NIR shoulder (675–690 nm). The severity of tungro also resulted in increasing first-order derivative values in the 550–580 nm ranges while decreasing in the 612–625 ranges. This can also be observed in the blast, increasing in 550–580 nm and 600–615 nm wavelengths compared to the susceptible control treatment. The changes suggest that the absorption in these ranges is affected by the disease infection.

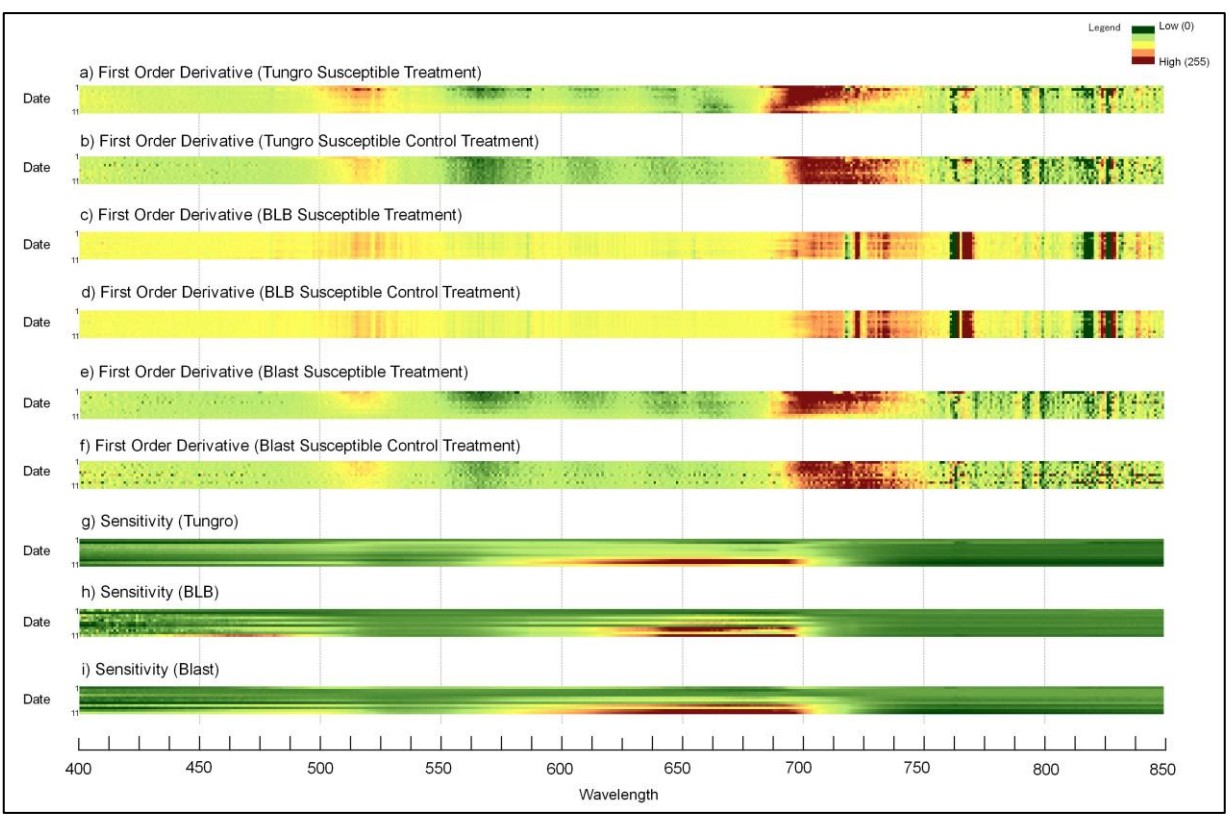

**Figure 7.** First-order derivatives of (**a**) tungro susceptible treatment, (**b**) tungro susceptible control treatment, (**c**) BLB susceptible treatment, (**d**) BLB susceptible treatment, (**e**) Blast susceptible treatment, (**f**) Blast susceptible control treatment; sensitivity ratio of (**g**) tungro, (**h**) BLB, and (**i**) Blast.

For BLB, the main difference between the susceptible (Figure 7c) and control susceptible treatments (Figure 7d) can be observed in the wavelength in the 680–700 nm range. The values at these wavelengths increased in the susceptible treatment, while the values in the control remained constant as the days passed (Day 1 to Day 11). This created a narrow shift in the unhealthy plant as the severity of the disease increased. Furthermore, a strong absorption band can also be seen at 653–655 nm in the susceptible treatment, which is absent in the control.

Five absorption ranges can be observed in the NIR region: 660–665 nm, 785–790 nm, 795–800 nm, 815–820 nm, and 830–833 nm. These absorption bands can be seen in all treatments and diseases.

On the other hand, the sensitivity ratio spectra enhance the differences between spectral signatures. It also determines sensitive and significant wavelengths for a disease. The ratio curve reflects both the change direction (increase or decrease) and the change magnitude of reflectance. Wider sensitivity can be observed in tungro from blue to red-edge ranges (400–725 nm), as indicated in a color change from Date 1 to Date 11 (Figure 7g). The wavelengths 400–560 nm (blue to green ranges) were sensitive at the onset of disease development; however, they became less sensitive as the disease severity increased. This was indicated by the change in color from yellowish to green color in these wavelengths. This is in contrast with the wavelengths from 570 to 710 nm, from yellow to red, which show increased sensitivity to tungro. In BLB, sensitivity can be shown in the 450–500 m (blue range) and 575–700 m (red–red-edge range) wavelengths suggested by increasing sensitivity ratio value (Figure 7h). Rice leaves inoculated by blast showed the same sensitivity as BLB, but they differ in a number of sensitive wavelengths. The blast susceptible variety had more sensitive wavelengths than BLB in the red–red-edge ranges, 560–710 nm (Figure 7i). A color change, from a lighter green to darker green, in the NIR region (i.e., >725 nm) was seen in blast and tungro.

### 3.4. Tungro, BLB, Blast Spectral Reflectance Resampling to the Different Optical Sensors

Sentinel 2A and 2B show a small difference in the spectral response based on the 2022 update; thus, separate resampling was carried out in these sensors. The study used 8 bands of Sentinel 2 (i.e., Band 1 to Band 8) for resampling since the spectral libraries only have 400 to 850 nm wavelengths. A small change to no significant change can be observed in band 1 to band 3 in all the diseases except in blast on the last date (Date 11). An increase in the resampled spectral reflectance was seen in band 4 (665 nm) and band 5 (705 nm), while band 6 (740 nm), band 7 (783 nm), and band 8 (842 nm) decreased for all the diseases under study over time for all the diseases.

In the case of Sequoia, it was observed that the red band (660 nm) increased over time. In contrast, green (550 nm), red-edge (735 nm), and NIR band (790 nm) decreased as the three diseases progressed.

Based on the resampling results, all diseases can be detected using optical sensors such as Sentinel 2 and Parrot Sequoia. Susceptible treatments in all the diseases had a significant change in resampled spectra as the disease severity increased. In contrast, the resistant and control treatments showed no significant change. However, the diseases under study were difficult to differentiate from each other. The three diseases had the same trend in each band (Figure 8).

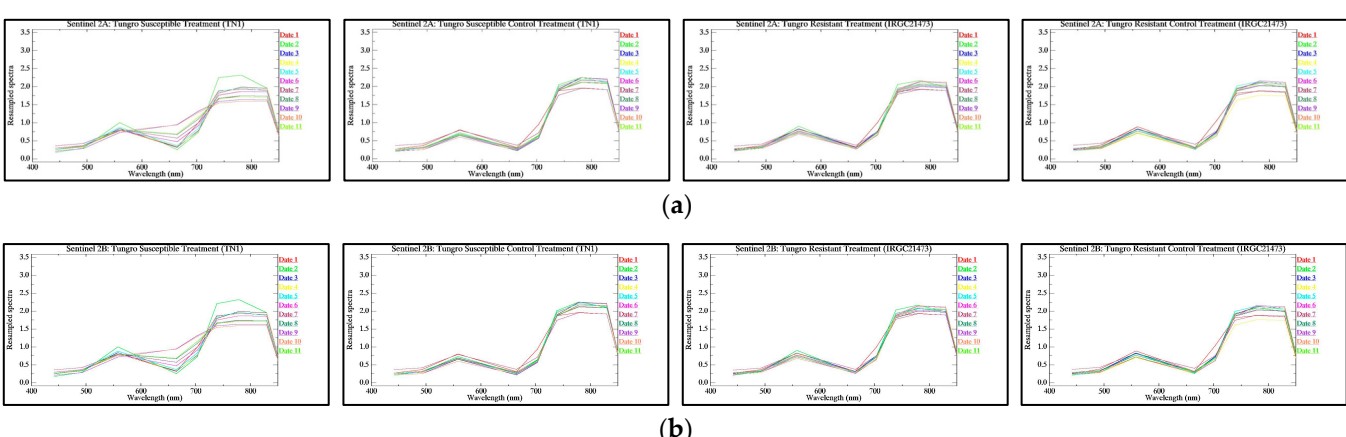

**Figure 8.** *Cont.*

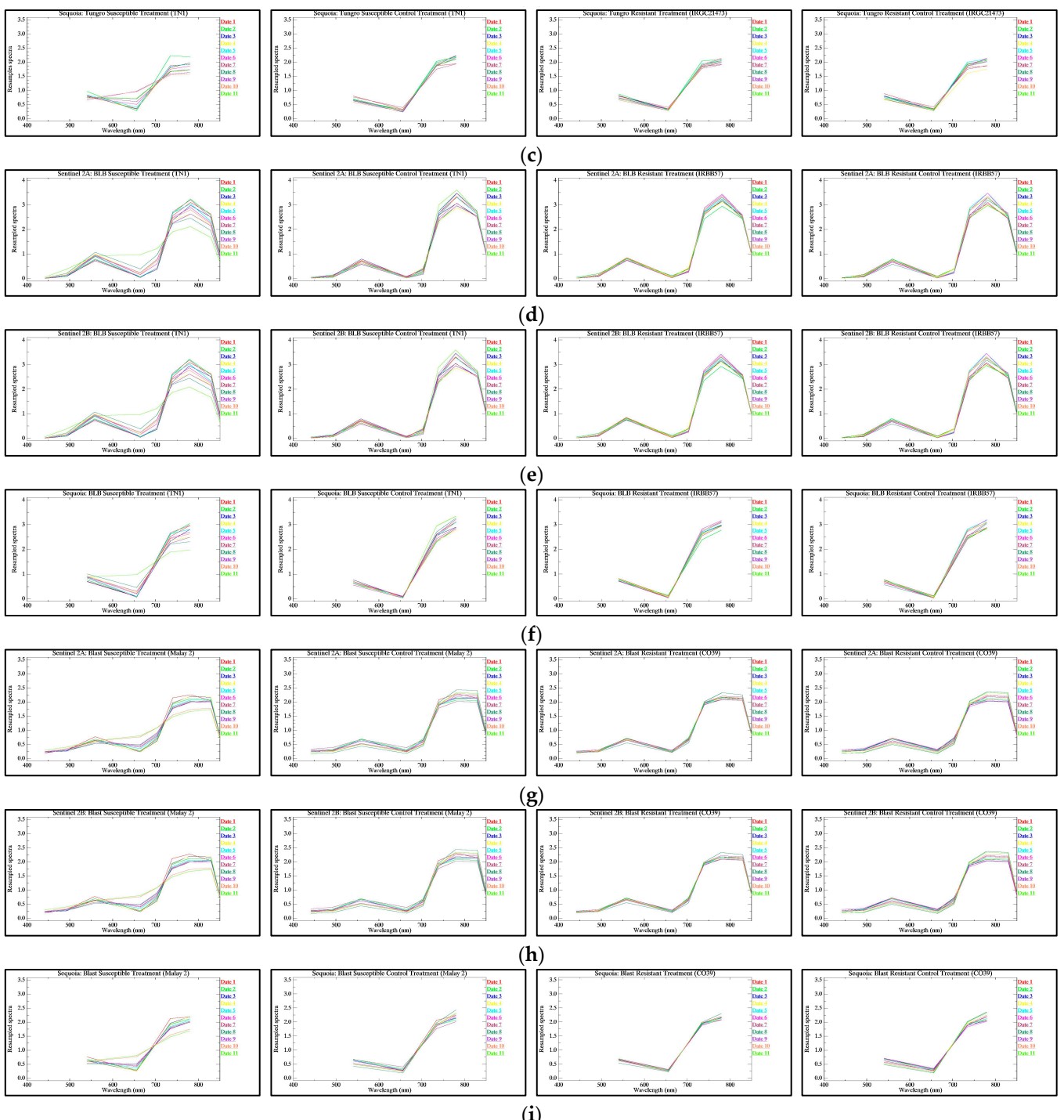

**Figure 8.** Resampled spectra of Tungro treatments (susceptible, control susceptible, resistant, and control resistant) to (**a**) Sentinel 2A, (**b**) Sentinel 2B, and (**c**) Parrot Sequoia. Resampled spectra of BLB treatments (susceptible, control susceptible, resistant, and control resistant) to (**d**) Sentinel 2A, (**e**) Sentinel 2B, and (**f**) Parrot Sequoia. Resampled spectra of BLB treatments (susceptible, control susceptible, resistant, and control resistant) to (**g**) Sentinel 2A, (**h**) Sentinel 2B, and (**i**) Parrot Sequoia.

### 3.5. Early Detection of Foliar Diseases Using Cluster Analysis Based on the Resampled Spectral Reflectance

Cluster analysis was performed to determine the earliest date to detect the three diseases. The different dates were grouped based on their similarities in resampled reflectance

to Sentinel 2A, 2B, and Parrot Sequoia. In Figure 9, the grouping of selected date numbers was emphasized to show the earliest detection of each disease.

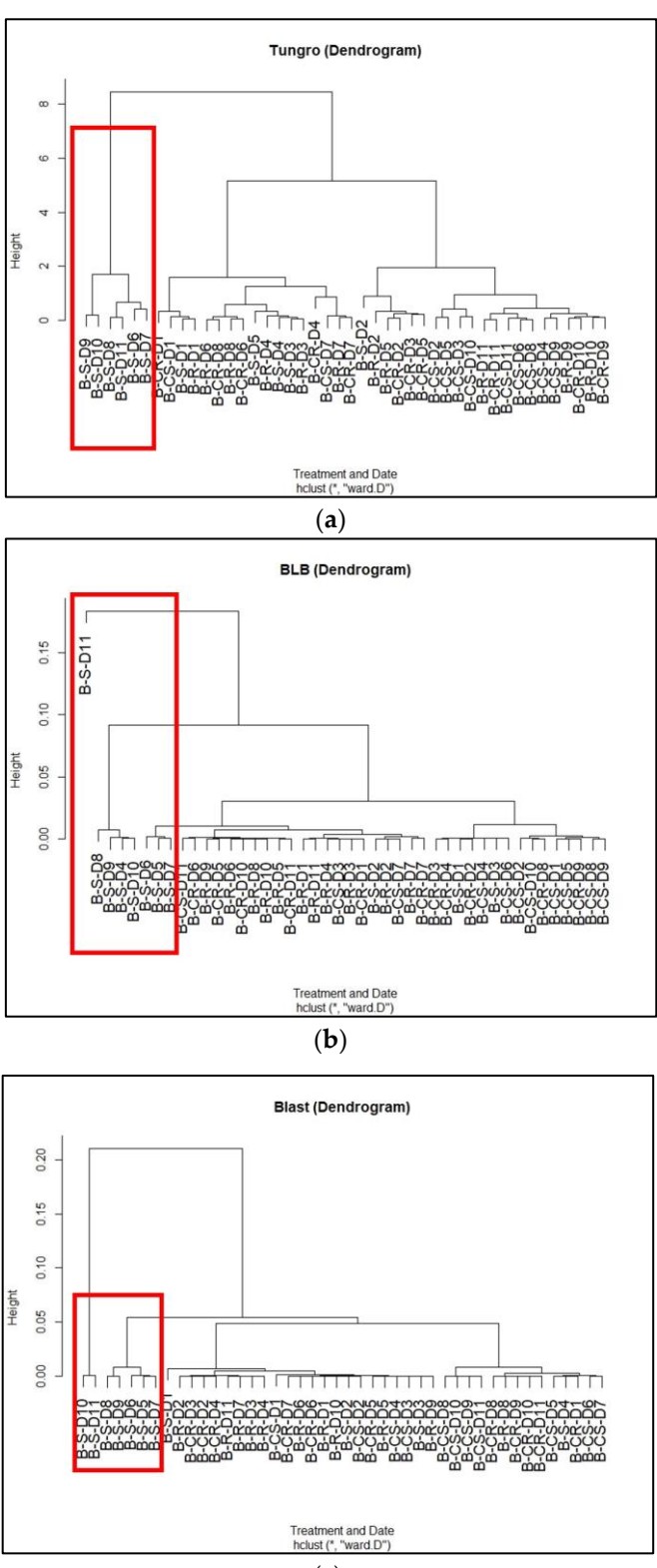

**Figure 9.** Hierarchal clustering of (**a**) tungro, (**b**) BLB, and (**c**) blast resampled reflectance. (* Hierarchical clustering agglomerative approach Ward's Method).

The main symptoms of rice tungro disease are stunting and manifesting yellow to orange discoloration of the leaves [30]. The results of cluster analysis showed that starting from date 6, 23 DAI, tungro can be detected. In this stage, the plants showed 11–30% height reduction and no distinct yellow to yellow-orange leaf discoloration (Figure 9a).

On the other hand, BLB can be detected starting at Date 4, 5 DAI (Figure 9b). Based on the SES scale for a field test, its score was 1, which means 1–5% of the leaf was covered by lesions. The clustering started from date 4 to date 11.

Clustering in blast shows that it can be detected in Date 5, 13 DAI (Figure 9c). The leaves have small round to slightly elongated, necrotic gray spots, about 1–2 mm in diameter, with a distinct brown margin (SES scale 3). A significant number of lesions are found on the upper leaves.

### 3.6. Partial Least Square Regression Analysis of Spectral Reflectance

Regression models were calculated to predict the disease severity in tungro, BLB, and blast (Table 4). The accuracy of the model for detecting the severity of the disease was evaluated using $R^2$ and RMSE. Higher $R^2$ values indicate a better fit between the model's predictions and the actual observations, while RMSE measures the model's prediction error compared to actual observed values.

**Table 4.** Overview of different regression model performances of each disease.

| Diseases | Parameters | | $R^2$ | | RMSE | | Prediction Accuracy (%) |
|---|---|---|---|---|---|---|---|
| | **Partial Least Square Regression** | | | | | | |
| | Ncomp [1] | | cal | val | cal | val | |
| Tungro | 6 | | 0.713 | 0.630 | 0.571 | 0.484 | - |
| BLB | 4 | | 0.634 | 0.289 | 0.645 | 0.670 | - |
| Blast | 2 | | 0.819 | 0.770 | 0.436 | 0.440 | - |
| | **Random Forest Regression** | | | | | | |
| | ntree [2] | | | | | | |
| Tungro | 1000 | | 0.858 | 0.617 | 1.170 | 1.428 | 71.43 |
| BLB | 100 | | 0.607 | 0.523 | 1.942 | 1.595 | 64.29 |
| Blast | 500 | | 0.773 | 0.575 | 1.447 | 1.772 | 76.92 |
| | **Support Vector Regression** | | | | | | |
| | Cost [3] | Gamma [3] | | | | | |
| Tungro | 10 | 0.001 | 0.916 | 0.815 | 0.897 | 0.815 | 78.57 |
| BLB | 100 | $1.00 \times 10^{-4}$ | 0.826 | 0.185 | 1.293 | 2.084 | 64.29 |
| Blast | 10 | $1.00 \times 10^{-4}$ | 0.859 | 0.830 | 1.140 | 1.121 | 73.08 |

[1] Number of components determined by using the one-sigma heuristic method in pls package selectNcomp. [2] ntree was determined from 100 to 1500 based on the lowest RMSE of the calibrated model. [3] Best parameters were determined using the function tune.svm() with 10-fold cross-validation error measurement.

Based on the leave-one-out cross-validation PLSR, the diseases need different numbers of components to minimize the number of factors for the model: 6, 4, and 2, for tungro, BLB, and blast, respectively. With two components, blast has the best-predicted model among the diseases with the highest $R^2$ ($R^2$cal = 0.819; $R^2$val = 0.770) and lowest RMSE (RMSEcal = 0.436; RMSEval = 0.440). In contrast, BLB has the lowest prediction model with low $R^2$ ($R^2$cal = 0.634; $R^2$val = 0.289) and high RMSE (RMSEcal = 0.645; RMSEval = 0.670).

Using Random Forest Regression, the different values of the parameter ntree used were 1000, 100, and 500 for tungro, BLB, and blast, respectively, to perform the analysis. This was optimized based on the lowest calibrated RMSE. Based on the results, tungro has the best-predicted model with the highest $R^2$ ($R^2$cal = 0.858; $R^2$val = 0.617) and lowest RMSE (RMSEcal = 1.170; RMSEval = 1.428) among the three diseases, while BLB has

the lowest prediction model with low $R^2$ ($R^2$cal = 0.607; $R^2$val = 0.523) and high RMSE (RMSEcal = 1.942; RMSEval = 1.595).

The parameters, cost, and gamma were defined using the function tune.svm() with 10-fold cross-validation error measurement. The best cost for tungro and blast is 10 while BLB is 100. BLB and blast have the same best gamma of $1.00 \times 10^{-4}$, while tungro has 0.001. Based on SVR, Tungro and blast have a better-fit model than BLB with $R^2$cal of 0.916 and 0.859 and RMSEcal of 0.879 and 1.140. BLB has a low predicted $R^2$val, 0.185, and a high RSME, 2.084.

## 4. Discussion

The study focused on three different diseases, tungro, BLB, and blast, caused by viruses, bacteria, and fungi. Depending on the pathogens, the three rice diseases were characterized by disease-specific symptoms in leaves. Tungro starts from leaf rolling to stunting and yellow to orange discoloration of leaves, while BLB shows water-soaked to yellow-orange stripes on leaf blades or leaf tips. Blast symptoms appear as lesions with elliptical or spindle-shaped, whitish-to-gray centers, and dark green to brownish borders.

Differences between the inoculated susceptible treatments from the resistant inoculated and control were noticeable in all the diseases (Figures 4–6). The development and severity of the diseases affected the spectral reflectance of rice leaves, as shown in Figures 4a, 5a and 6a. Reflectance in red and red-edge ranges of the susceptible treatment was increasing as the diseases progressed. Both tungro and blast increased in the 600 to 700 nm wavelengths, while BLB increased in 550 to 680 nm (Table 5). This can be attributed to the strong reflectance of pigments in the red and red-edge ranges. Pigments such as chlorophyll-a and b and carotenoids decrease as stress (biotic and abiotic) is introduced to the plants [31,32]. There are no symptoms that can be seen in the inoculated resistant treatment of all three diseases. With this, the spectral reflectance of resistant treatment and control exhibit the same trends.

**Table 5.** Summary table of the diseases' (i.e., tungro, BLB, and blast) effects on the different spectral ranges.

| Spectral Ranges | Tungro | BLB | Blast |
|---|---|---|---|
| Violet to Blue | Increased in 400–500 nm | No change | No change |
| Green | No change | Increased in 550–580 nm (Green-Yellow) | No change |
| Red | Increased in 600–680 nm | Increased in 580–680 nm | Increased in 600–675 nm |
| Red-edge | Shift in 675–690 nm | Shift in 680–705 nm | Shift in 675–700 nm |
| NIR | Decreased | Decreased | Decreased |

The three diseases' severity resulted in different red-edge inflection points: tungro in 675–690 nm, BLB in 675–705 nm, and blast in 675–700 nm wavelengths (Table 5). These regions became less steep and shifted into shorter wavelengths compared to the resistant and control treatments (Figures 4b–d–6b–d). In comparison, the region in healthy plants shows a sharp increase in reflectance and remained constant throughout the disease progression. The red-edge region was defined as the inflection point position on the slope connecting the red and NIR spectral regions. It featured the transition from visible range strong absorption by leaf chlorophyll to the structurally dominated reflectance in the NIR. This has been identified as sensitive to changes in chlorophyll content. With high chlorophyll content, the shoulder shifts toward the longer wavelengths, while it shifts toward the shorter wavelengths with low chlorophyll content [33,34].

Particularly in the later stages of the diseases, susceptible varieties have lower NIR reflectance values than other varieties. Because absorptance in this area was controlled by interior leaf structure, it rapidly declined as the disease severity increased. The ratio

of the mesophyll cell surface to the intercellular air gaps in this area controls reflection, which is generally high. Reflectance was also high due to refractive discontinuities between intercellular air spaces and cell walls [33]. However, these leaf structures were damaged by the disease during its pathogenesis.

In violet to blue ranges (400–550 nm), reflectance had a small change for all the diseases. However, a significant increase can be observed on the date that the SES scale is 9, and the diseases were in their late stages. There is overlapping absorption of chlorophyll and carotenoids; thus, we had difficulty in estimating the effect of the diseases in these ranges. Only during the leaf senescence, in which chlorophyll breaks down before carotenoid degradation, does the carotenoid-to-chlorophyll ratio increase [34].

Compared to tungro and BLB, blast has a weaker response in the early development of the disease. This can be due to the symptom progression. Early infection of blast has small lesions and occupies a small area in the leaf, making it hard to detect during the early onset of the disease [21]. Thus, the spectral reflectance in early dates has a high mixture of healthy and diseased spectra. It is also shown that blast is sensitive in 675–700 nm wavelengths. The same result was identified in the study of [15] wherein 600–700 nm and 720–1000 nm wavelengths were highly sensitive to leaf blast infection. However, the acquisition of this study was carried out daily to assess the response of spectra to the progression of the disease.

Singh et al. (2012) also studied the use of hyperspectral data to detect BLB at different disease severity levels, which showed notable differences between healthy and diseased rice plants in the NIR region (770–860 nm and 920–1050 nm) [19]. This finding is consistent with the results of this study; the red–red-edge region (575–700 nm) was the most sensitive region to BLB infection. A notable increase can be observed in 550–580 nm wavelengths, the green to yellow region of spectra (Table 5). This can be attributed to the symptom of BLB; the green leaves turn yellow to straw-colored and wilt as the disease progresses [7].

First-order derivatives show the wavelength ranges in which the diseases influenced the light reflection and absorption. Low values in first-order derivatives show low reflection and high absorption in the wavelengths. In tungro and blast, absorption bands show changes in the range of 550–580 nm and 700–750 nm. The change shows increasing absorption in these wavelengths as the severity of the disease increases. The green range (550–580 nm) change was due to the degradation of chlorophyll in the plants. Chlorophyll molecules absorb solar radiation and convert it into stored chemical energy for plants [33,34]. It strongly absorbs red and blue regions but reflects green range wavelengths. Thus, degrading chlorophyll decreases the absorption in these ranges. Moreover, tungro caused chloroplasts in the mesophyll cells to disintegrate, thus decreasing photosynthetic and accessory pigments in affected tissues [35].

Moreover, tungro infection shows decreasing absorption in 612–615 nm wavelengths. The change can be attributed to the symptom exhibited by tungro infection; leaves have a distinct yellow to yellow-orange color then turn brown as the leaves die in late stages. This transition in leaf color results in a corresponding increase in reflectance in the red–red-edge range [30].

A high sensitivity ratio of susceptible treatments is shown in red and red-edge wavelengths. These ranges have been identified as sensitive to changes in the chlorophyll content of the plant, especially in the red-edge region. Chlorophyll and anthocyanin are reflected strongly in red wavelengths.

Resampling the spectral libraries to Sentinel 2 and Parrot Sequoia spectral functions shows disease detection, especially at the plant leaf level. The spectral signatures can be distinguished in the different bands. Red and NIR band values changed as the three diseases progressed. The red-edge bands present in both sensors can provide key information on the vegetation state. This could be the reason for possible disease detection.

Cluster analysis was used to detect the earliest stages of the three diseases as the dates with similar values were grouped. Based on the results, the diseases can be detected in different SES scales. Tungro can be detected when plant height is reduced to 11–30% and

when no distinct yellow to yellow-orange leaf discoloration is observed. This was seen at 23 DAI, which has an SES score of 5. In contrast, BLB was detected at 5 DAI, in which the area affected was less than 5%. Blast was detected when the lesions were mostly found on the upper leaves (SES scales of 3 or higher). These lesions are small roundish to slightly elongated with necrotic gray spots and a distinct brown margin.

Three regression techniques were used in the study to predict the three foliar diseases: Partial Least Square, Random Forest, and Support Vector Regression. RFR and SVR are examples of machine-learning techniques that use algorithms for model prediction and classification [29]. The results showed that PLSR for the three diseases has the lowest prediction models with RSMEval of 0.484, 0.670, and 0.440 for tungro, BLB, and blast, respectively. SVR models resulted in the highest $R^2$val for tungro and blast, at 0.815 and 0.830, respectively. However, the SVR $R^2$val was the lowest for BLB among the three regression models. Higher $R^2$ values represent smaller differences between the observed data and the fitted values. The validation results of the PLSR and SVR models in tungro and blast show accuracy levels that are promising to be used in estimating the severity of the disease in leaves, whereas RFR shows the best results for BLB.

## 5. Conclusions

The study aimed to detect and differentiate three major diseases (i.e., blast, BLB, and rice tungro) in the Philippines by performing a multi-temporal spectral analysis. Reflectance spectra of rice leaves infected with rice tungro disease, BLB, and blast were affected by each disease, resulting in different spectral signatures. It assessed the potential of satellite and airborne optical sensors (e.g., Sentinel 2 and Parrot Sequoia) for the early detection and discrimination of the diseases using these spectral features. Early detection and monitoring of these diseases may help in the proper and timely implementation of disease management and interventions.

Based on the results, the red and red-edge ranges were the most sensitive to the three diseases. The reflectance increased in both ranges as the disease incidence and severity progressed. The diseases created different red-edge inflection points: tungro in 675–690 nm, BLB in 675–705 nm, and blast in 675–700 nm wavelengths. In comparison, NIR reflectance also decreased as the three diseases progressed. A more pronounced difference in NIR reflectance between healthy and unhealthy plants can be seen at the later stages of the diseases.

The study shows that the reflectance in the 400–850 nm range is governed by the biochemical and physical properties of the leaves, including leaf pigmentation, water, and internal and external leaf structures. However, no procedure was carried out to determine the levels of change in the biochemical and physical properties of the diseased leaves. Further research is recommended to assess these properties, along with spectral measurements, as the diseases progress.

Spectral analysis such as reflectance sensitivity and first-order derivatives in leaves can show the differences between each disease and the disease's progression. In addition, resampling the spectral reflectance to sensors' spectral function (i.e., Sentinel 2 and Parrot Sequoia) can possibly detect the diseases. However, these three diseases cannot be discriminated by using these optical sensors within these ranges alone. Combining optical sensors with other sensors that are sensitive to thermal band ranges (10.8 μm to 12 μm) may have the potential to further differentiate these diseases. Additionally, measurements and analysis of the spectral reflectance demonstrated the possibility of disease detection based on the changes in the spectral reflectance ranges.

The three regression techniques can demonstrate the estimation of disease severity using leaf spectral reflectance. Assessment of the disease severity can help in the monitoring and implementation of proper disease management in rice diseases such as tungro, BLB, and blast.

**Author Contributions:** Conceptualization, J.R.F.M. and E.C.P.; Formal analysis, J.R.F.M.; Writing—original draft, J.R.F.M.; Writing—review & editing, M.Y. (Megumi Yamashita) and E.C.P.; Supervision, M.Y. (Megumi Yamashita), M.Y. (Mitsunori Yoshimura) and E.C.P. All authors have read and agreed to the published version of the manuscript.

**Funding:** This research was a part of Ms. Mirandilla's master's thesis under the scholarship of the Department of Science and Technology – Engineering Research and Development for Technology (DOST-ERDT).

**Data Availability Statement:** The data that support the findings of this study are available by request from the corresponding author.

**Acknowledgments:** We acknowledge the Department of Science and Technology-Engineering Research and Development for Technology (DOST-ERDT) and the Philippine Rice Research Institute (PhilRice) for providing support to the first author.

**Conflicts of Interest:** The authors declare no conflict of interest.

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
