# Peer review of "Leaf Spectral Analysis for Detection and Differentiation of Three Major Rice Diseases in the Philippines"

_remotesensing, doi:10.3390/rs15123058_

Round 1

Reviewer 1 Report

This work conducts an experiment with three kinds of diseases over rice leaves, by splitting the samples into susceptible and resistant groups. The height and reflectance of leaves is measured over 37 days, and the results were compared according to the reflectance, their first-order derivatives, the clustering of daily samples and a small prediction model.

The experiment has been well conducted, and my main concerns are about the processing of the collected reflectance signatures. Firstly, it is explained that several measurements are replicated per sample (4 x 15). How were these measurements aggregated? I think it is not explained. For instance, it could be relevant to know whether the measurements may change on the front and back of leaves, or if there existed noise samples that were discarded during the aggregation.

The collected spectral signatures were downsampled with the wavelengths present in satellite and UAS-based multispectral sensors. I cannot understand what is the purpose of this; the main advantage of the spectrometer is that it collects many spectral samples, in comparison with multispectral sensors. Therefore, signatures are downsampled to a few wavelengths that may not be the most relevant features to discern infected and healthy leaves. Another approach would be to extract the most relevant features, according to the weight of every wavelength in the classification of healthy and unhealthy leaves. Otherwise, the spectral signatures can also be transformed into a space of lower dimensionality which is more representative, regardless of the label of each sample (e.g., using Factor Analysis). Then, the transformed features could be clustered to discern whether the learned space is relevant to this classification problem. 

Note that these are simply suggestions from my own experience, and I do not intend to make you include these methods. However, the conversion of hyperspectral sensors into multispectral sensors is hard to understand. You have the best possible data for this classification task.

Also, the different states of leaves are compared according to their first-order derivative, which is known to be independent on the signature scale. For instance, it has been used to classify hyperspectral signatures since the shape of signatures from the same material is considered to remain similar, despite having different magnitudes due to the viewing angle, light, etc. From the illustrated signatures, it seems that the main difference is the magnitude of the signatures rather than the shape. For instance, there can be observed very different shapes within a case study such as the Tungro susceptible treatment, but these are found in the same sample group, not between different groups.

Finally, the classification could be pushed a bit further by testing some Machine Learning algorithms to check how easy is to discern healthy and infected samples. I have read that you have used R, and maybe it could be easy to find some ML libraries. I do not expect you to describe every ML algorithm; showing the results of different algorithms in a table could be enough. Also, the overall accuracy is a better indicator for readers, especially those who are not immersed in ML.

I have attached the document with some highlighted words from either incomplete phrases or wrong words.

Author Response

Thank you

Reviewer 2 Report

This paper examined the spectral characteristics of plant diseases using Sentinel-2 images, which is a very interesting topic but needs some modifications regarding methodologies and interpretations for future consideration of publication.

Line 222. Please present the reason the PLSR has advantages for hyperspectral image analysis.

Line 397. Instead, RFR (random forest regression) or SVR (support vector regression) could provide more accuracy because they can handle the non-linear aspects of the relationships. Can you present the quantitative comparisons between the several regression methods?

Line 453. The increase in red reflectance and the decrease in NIR reflectance seem reasonable. However, the increase in green reflectance for BLB looks quite strange. Please provide more discussions about it.

Author Response

Thank you.

Round 2

Reviewer 1 Report

Dear authors,

Thanks for addressing my previous concerns. However, I have a few comments on the revised article.

First, it is concluded that the disease symptoms are visible even narrowing the wavelength interval to the ones in Parrot Sequoia and Sentinel satellites. However, the plots only show the line chart for the susceptible treatments, instead of comparing susceptible and resistant.

On the other hand, I still think the overall accuracy should be annotated regarding the prediction models as this metric is easier to interpret. Similarly, a summary of the dataset (e.g., the number of samples for each kind of treatment) would help the reader to better understand the prediction results. This way we can evaluate whether the configuration of RF, for instance, is adequate. The number of employed trees is too high for the size of the dataset that (I guess) you are using. 

Author Response

Thank you.
